# Isolation and Identification of a Novel Phlebovirus, Hedi Virus, from Sandflies Collected in China

**DOI:** 10.3390/v13050772

**Published:** 2021-04-27

**Authors:** Ziqian Xu, Na Fan, Xin Hou, Jing Wang, Shihong Fu, Jingdong Song, Mang Shi, Guodong Liang

**Affiliations:** 1State Key Laboratory of Infectious Disease Prevention and Control, Key Laboratory for Medical Virology, National Institute for Viral Disease Control and Prevention, Chinese Center for Disease Control and Prevention, Beijing 100052, China; xuzq@ivdc.chinacdc.cn (Z.X.); fanna0925@163.com (N.F.); shihongfu@hotmail.com (S.F.); sjdccdc@163.com (J.S.); 2The Center for Infection & Immunity Study, School of Medicine, Sun Yat-sen University, Guangzhou 510006, China; houx5@mail2.sysu.edu.cn (X.H.); wangj796@mail2.sysu.edu.cn (J.W.)

**Keywords:** phlebovirus, sandfly-borne phleboviruses, *Phlebotomus chinensis*

## Abstract

We report the isolation of a newly recognized phlebovirus, Hedi virus (HEDV), from *Phlebotomus chinensis* sandflies collected in Shanxi Province, China. The virus’ RNA is comprised of three segments. The greatest amino acid sequence similarity of the three gene segments between this virus and previously recognized phleboviruses is 40.85–63.52%, and the RNA-dependent RNA polymerase (RdRp) amino acid sequence has the greatest similarity (63.52%) to the Rift Valley fever virus (RVFV) ZH-548 strain. Phylogenetic analysis of the amino acid sequence of the virus RdRp indicated that HEDV is close to RVFV and distinct from other phleboviruses, forming its own evolutionary branch. We conclude that it is necessary to increase the monitoring of phleboviruses carried by sandflies in China.

## 1. Introduction

*Phlebovirus* is a genus of RNA viruses belonging to the family *Phenuiviridae* in the order *Bunyavirales*. According to the latest classification of the International Committee on Taxonomy of Viruses (ICTV), the genus contains 60 species [1]. These viruses are mainly distributed in the New World tropics and Old World semi-arid and temperate regions, including the Mediterranean, North Africa, and central and western Asia [2]. Phleboviruses are mainly transmitted by sandflies, although a few are transmitted by mosquitoes (such as Rift Valley fever virus) and ticks (such as Mukawa virus) [1]. The new classification places two well-known tick-borne phleboviruses, severe fever with thrombocytopenia syndrome virus and heartland virus [3,4], in the new genus *Bandavirus* [1].

Rift Valley fever virus (RVFV) is the most well-known phlebovirus. It is a zoonotic pathogen transmitted by mosquitoes that can cause disease and death in a variety of ruminants (sheep, goats, cattle, etc.) and fever, blindness, encephalitis, death and other signs and symptoms in humans [5,6,7]. The virus is prevalent in parts of Africa, but has spread to the Arabian Peninsula and is likely to spread further [8].

In this study, during a surveillance of viruses carried by sandflies, a novel phlebovirus, Hedi virus (HEDV), was isolated from sandflies collected in Shanxi Province, central China. The virus is a close relative of RVFV based on the phylogenetic analysis of the RdRp gene of this virus.

## 2. Materials and Methods

### 2.1. Sample Collection

In June 2018, a routine surveillance of viruses carried by sandflies was conducted in Shanxi Province (112°5′–114°4′ E, 37°40′–38°31′ N), central China [9,10]. Sandflies were captured in light traps (Wentaitai MM200 traps, Guangzhou Changsheng Chemical Technology Service Co., Ltd., Guangzhou, China) and transported to the laboratory in Beijing on dry ice. The specimen collection time was 18:00 p.m. to 6:00 a.m. on the following day [11].

### 2.2. Virus Isolation and Identification

All specimens were classified according to morphology, collection time, and collection site. The sandfly specimens were ground in batches (50–100 sandflies/batch), centrifuged (4 °C, 12,000 rpm, 30 min), and 100 μL of the supernatants were inoculated into BHK-21 (baby hamster kidney) and C6/36 (*Aedes albopictus*) cells for virus isolation [12]. Phlebovirus-specific primers targeting 554 nt of the L gene [13] were used to search for phleboviruses in the supernatants after three subcultures.

### 2.3. Virus Genome Sequencing

To study the molecular characteristics of HEDV, we used next-generation sequencing (NGS) and reverse transcription PCR to determine its complete gene sequence. Briefly, the supernatants of BHK-21 cells infected with 2nd-passage HEDV were collected. After extracting total RNA with the QIAamp Viral RNA Mini Kit (Qiagen, Hilden, Germany), a library was constructed using the TruSeq total RNA library approach (Illumina, San Diego, CA, USA) and sequenced on the HiSeq 4000 platform (Illumina). After the low-quality reads were removed from the sequencing data using Trimmomatic, *de novo* assembly was performed using the Trinity program [14]. The assembled contigs were compared to a non-redundant protein database (nr) using blastx, and contigs related to the three phlebovirus segments were identified. Based on the assembled phlebovirus-related contigs, Primer 5.0 was used to design 23 pairs of primers covering the three gene segments of the virus (Appendix A), and PCR amplification and nucleotide sequencing of the HEDV gene sequence were performed. Conserved primers corresponding to the ends of the phlebovirus sequence were used to amplify the 3′ and 5′ non-coding regions of the virus gene [15]. The entire genome sequence of HEDV has been registered in the GenBank database (accession numbers MW368831–MW368833).

### 2.4. Phylogenetic Analysis

For phylogenetic analysis, the amino acid sequences of the RdRp, GPC, N, and NSs proteins of HEDV were compared with those of 59 phleboviruses (at least 1 virus strain was selected for each species for analysis; the Mariquita virus sequence was not available) (Appendix A). The amino acid sequences were aligned using MAFFT (7.450-win) [16]. The 5′ and 3′ terminal sequences were removed manually, and then trimAl was used to prune the sequences [17]. PhyML (version 3.1) [18] was used to construct the phylogenetic tree using the maximum likelihood method, LG amino acid substitution model, and SPR tree topology optimization algorithm.

### 2.5. Electron Microscopy

Negative-stained electron-microscopic specimens were prepared using infected BHK-21 cell supernatants (passage 4) mixed 1:1 with 2.5% paraformaldehyde, fixed onto Formvar/carbon-coated grids, and stained with 3% phosphotungstic acid (pH 6.3). The viral particles were then observed using a transmission electron microscope 109 (TECNAI 12, FEI, Blackwood, NJ, USA) with an acceleration voltage of 80 kV [19].

### 2.6. Growth Curve of Isolated Virus

BHK-21 cells at 80% confluency were inoculated with 100 μL suspension of virus stock and cultured in a humidified incubator at 37 °C and 5% CO_2_ [12]. One tube was collected every 24 h after infection for 21 days and stored at −40 °C. Subsequently, a quantitative RT-PCR (qRT-PCR) assay was designed for specific detection of the newly isolated virus in the polyprotein gene. The primers consisted of MF (TGCATCACCATTCCAACCGA), MR (CTCTGCTCAAGAAGGCACCA), and MP (FAM-TATGACCCAGCAGAATAATAGGCAGAGC-TAMRA). The virus RNA was determined with the qRT-PCR assay and a virus proliferation curve was generated.

### 2.7. Species Identification of Sandfly Specimens

According to the instructions of the DNA Extraction Kit (QIAamp; Qiagen, Valencia, CA, USA), DNA was extracted from sandfly specimens from which the virus was isolated. PCR was used to amplify the cytochrome c oxidase I (COI) gene using primers (LCO1490: GGTCAACAAATCATAAAGATATTGG, and HCO2198: TAAACTTCAGGTGACCAAAAAATCA) [20]. The PCR product was sequenced by Sanger sequencing and was compared to sequences at the NCBI website using the Basic Local Alignment Search Tool (BLAST, https://www.ncbi.nlm.nih.gov/blast (accessed on 22 October 2020)).

## 3. Results

### 3.1. Isolation of Hedi Virus

A total of 3996 sandfly specimens were collected. Detection of the arthropod cytochrome c oxidase I gene and analysis of sandfly specimen 1867-2 showed that this batch of sandflies was *Phlebotomus chinensis*. The specimens were inoculated into BHK-21 and C6/36 cells for virus isolation in 51 batches. All of the C6/36 cell supernatants were phlebovirus-negative, while 11 (21.6%) of the BHK-21 cell supernatants were positive. After sequencing the PCR products, BLASTX queries revealed that 1 (sample no. 1867-2) of the 11 positive samples was similar to RVFV (71.2%), while the other 10 were similar to Wuxiang virus [11,21]. Surprisingly, although the phlebovirus sequence was detected in the BHK-21 cell culture supernatant of the 1867-2 specimen, the sample did not show cytopathic effects (CPE) in BHK-21 cells. After five passages, the virus sequence was still detected, but no CPE was seen, indicating that this virus can be amplified in BHK-21 cells, but does not cause CPE in them. We named this virus isolate Hedi virus 1867-2 (HEDV1867-2).

### 3.2. Genome Sequence Characterization of Hedi Virus

The complete genome sequence of HEDV1867-2 was recovered by NGS and confirmed with reverse transcription PCR. The genome of the HEDV1867-2 isolate had three segments: the large (L) segment (6412 nt) encoding RNA-dependent RNA polymerase (RdRp); the medium (M) segment (4408 nt) encoding polyprotein; and the small (S) segment (1771 nt) encoding nucleoprotein (N) and a non-structural protein (NSs) (Figure 1).

The 2093 amino acids encoded by the 6282-nt open reading frame (ORF) of the L segment of HEDV1867-2 were most similar to the RdRp protein (YP 003848704) of RVFV strain ZH-548 (Egyptian human isolate from 1977) (63.52%) (Figure 1). Analysis based on sequence alignment showed that the HEDV1867-2 L segment possessed the conserved palm motifs Pre-A, A through E, and G of the phlebovirus L segment, and the conserved N-terminal motifs H...D...PD...ExT...K of the phlebovirus endonuclease domain (Appendix A) [22,23,24,25].

The 1386 amino acids encoded by the 4161-nt ORF deduced for the HEDV1867-2 M gene were 40.85% (coverage 99%) similar to the glycoprotein precursor protein (GPC) (AHK60932) of the Karimabad virus strain I-58 (Iranian sandfly isolate from 1959), and with RVFV (ZH-548) was 44.83%(coverage 74%). After translation, the deduced GPC ORF was divided into glycoproteins Gn (530 aa) and Gc (468 aa), and non-structural protein NSm (193 aa). The S gene contains two ORFs, of which the 247 amino acids encoded by the 3′ ORF (744 nt) were 57.20% similar to the N protein (AEB70977) of Durania virus Co Ar 171162 (Colombian sandfly isolate from 1986) and 56.44% similar to that of RVFV (ZH-548); the 286 amino acids encoded by the other ORF (861 nt) were 44.40% similar to the NSs protein (AHK60936) of Gabek Forest virus (Sud AN 754-61 from a Sudanese African spiny rat isolate from 1961) and 35.18% similar to RVFV (ZH-548) (Figure 1).

### 3.3. Phylogenetic Relationship of HEDV

The RdRp gene of HEDV1867-2 formed a monophyletic cluster with RVFV (Figure 2A). On the other hand, the viral GPC gene was clustered with those of Karimabad virus and Sicilian sandfly fever phleboviruses (Dashli virus, Toros virus, and sandfly fever Turkey virus) isolated from sandflies (Appendix A); the viral N protein gene was related to Tapara virus (Brazilian sandfly virus isolate) (Appendix A); the viral NSs protein gene was adjacent to Karimabad virus (Appendix A). The L gene of HEDV1867-2 was divergent from all currently known variations of RVFV, represented by 253 known strains (Figure 2B). These results suggested that HEDV1867-2 is a phlebovirus with a potentially close evolutionary relationship to RVFV.

### 3.4. Viral Morphology and Replication

Using electron microscopy, 100-nanometer-diameter round virus particles with a membrane and spike structure were seen in the supernatant of BHK-21 cells infected with HEDV1867-2 (fourth passage). To characterize the replication of HEDV1867-2 in BHK-21 cells, cell supernatants were harvested on days 1 to 21, and viral genome amplification was detected. The viral load peaked at day 11 and then began to decline (Appendix A).

To investigate whether HEDV1867-2 can cause CPE in other mammalian cell lines, Vero and Vero E6 (African green monkey kidney) cells were inoculated with the supernatant of HEDV1867-2-infected BHK-21 cells and passaged twice. No CPE was seen in the two virus passages in either cell line. No HEDV1867-2 genes were detected in the supernatants of cultures of the two cell lines. Therefore, HEDV1867-2 did not replicate in monkey kidney Vero cells or in Vero E6 cells.

## 4. Discussion

The phlebovirus RdRp is the core protein that is associated with the replication of the virus genome. If the difference in the RdRp amino acid sequence of phleboviruses exceeds 5%, viruses are considered distinct [26]. The RdRp amino acid sequence of HEDV1867-2 had the highest homology with RVFV among 60 phleboviruses, at 63.52%. The amino acid length of the L segment coding region of HEDV1867-2 was only one residue longer than that of RVFV (2093 vs. 2092 aa). In the phylogenetic analysis of the viral RdRp gene, HEDV1867-2 and RVFV formed a monophyl, but are highly divergent from each other, suggesting that they have taken a long time to evolve to form their respective virus RdRp proteins (Figure 2). Although the data here are far from enough to depict the complete evolutionary history of this important group, the divergent nature of HEDV suggests that the virus is a novel species, one currently most similar to RVFV. This is the first time that a similar virus has been isolated from sandflies.

Although phylogenetic analysis indicates that HEDV and RVFV have an evolutionary relationship, there are marked differences in the infectivity of the two viruses for tissue culture cells. RVFV causes CPE in a variety of tissue culture cells, including BHK-21, Vero, and Vero-E6 cells [27]. Although it did not cause significant CPE in mosquito C6/36 cells, C6/36 cells amplified RVFV [28]. However, HEDV1867-2 replicated only in BHK-21 cells, with no CPE. Moreover, it did not replicate or cause CPE in monkey kidney Vero and Vero-E6 cells or mosquito C6/36 cells, suggesting that HEDV has a narrow host range. Further research should examine whether HEDV can cause lesions or replicate in other cell cultures.

While collecting sandflies, 74 mosquitoes were collected at the same site, including 52 Culex pipiens pallens, 13 Armigeres subalbatus, 8 Aedes, and 1 Anopheles [21]. The collected mosquitoes were separated into three groups and inoculated in parallel into BHK-21 and C6/36 cells, which were passaged three times. However, no CPE was seen in the cell cultures, and the HEDV1867-2 genome was not detected in the culture supernatants, suggesting that HEDV does not circulate in the local mosquitoes. Combined with the finding that HEDV1867-2 cannot be amplified in C6/36 cells, this suggests that HEDV is transmitted by sandflies, not mosquitoes, unlike RVFV, which is transmitted by a variety of mosquitoes, including Culex tritaeniorhynchus Giles and Culex quinquefasciatus [6]. So far, there has been no report of RVFV isolated from sandflies.

## 5. Conclusions

In this study, the newly recognized phlebovirus, Hedi virus, was isolated from sandflies collected in the field. The genomic structure, nucleotide and amino acid sequence homology, and molecular genetics evolution analyses of the virus indicated that it is a virus related to RVFV, which is transmitted by mosquitoes. This is the first time that a close relative of RVFV has been found in sandflies. However, we report only the isolation and identification of the virus. Further studies should examine the host and epidemic range of the virus in the field, especially whether the virus can infect humans, livestock, and other animals. Therefore, it is of great public health importance to strengthen the detection and monitoring of similar viruses in sandflies, including seroepidemiological studies of the virus in local people, livestock, and other animals.

## Figures and Tables

**Figure 1 viruses-13-00772-f001:**
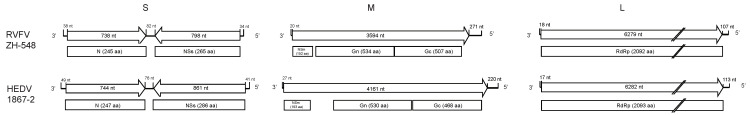
Schematic of the genome organization of HEDV1867-2 and Rift Valley fever virus strain ZH-548. Lines represent gene segments; arrows represent coding regions; and boxes represent transcribed proteins. The full lengths of the three gene segments of HEDV1867-2 and the deduced ORFs are 3 to 567 nt longer than those of ZH-548. The amino acids encoding the HEDV1867-2 proteins, except for glycoproteins Gn and Gc, which are 4 aa and 39 aa shorter, respectively, were 1 to 41 aa longer than those of ZH-548, and the non-structural proteins NSs and NSm were 21 and 41 aa longer, respectively.

**Figure 2 viruses-13-00772-f002:**
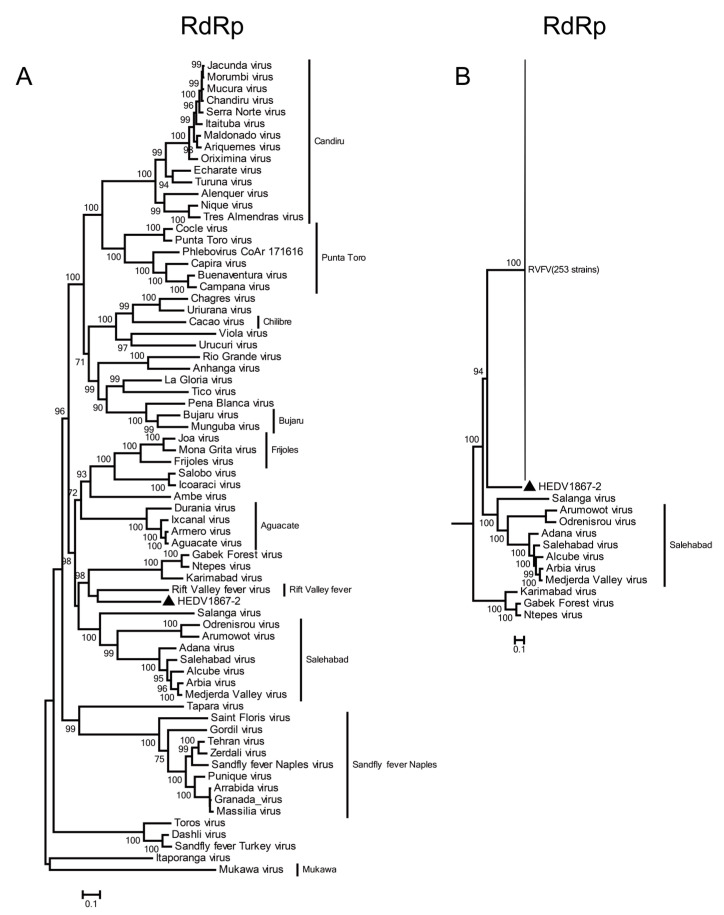
The evolutionary relationship between the RdRp amino acid sequences of HEDV1867-2, (**A**) 59 species of phleboviruses and (**B**) 253 strains representative of the RVFV diversity. The HEDV1867-2 virus identified in this study was marked with black triangles. The trees were reconstructed with maximum likelihood approach.

## Data Availability

All of the materials and data that were used or generated in this study are described and available in the manuscript and Appendix A.

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
