# Peer review of "Isolation and Identification of a Novel Phlebovirus, Hedi Virus, from Sandflies Collected in China"

_viruses, 2021, doi:10.3390/v13050772_

Round 1
Reviewer 1 Report
This is a paper reporting the discovery and characterization of a novel phlebovirus from sandflies. The authors argue convincingly that based upon sequence analysis, the virus is sufficiently distinct to warrant taxonomic designation as a new species. They also present compelling need for continued surveillance of phleboviruses in sandflies (in China).
While the title is quite clear, the introduction of the paper is unnecessarily obscure. The third paragraph appears to have been truncated and fails to state the purpose of the paper and the key finding. It should be clearly stated that the study is meant to report on a NOVEL virus (SBV1867-2) and introduce the purpose of the report.
That this is a rift valley fever like virus found in sandflies (rather than mosquitoes) is an important finding. The authors indicate that sequence difference infer that the divergence of mosquito and sandfly phleboviruses “have taken a long time to evolve”. It would be nice if they set a clock to this divergence and/or look at the chronology of bunyavirales divergence to determine whether it is likely that this divergence corresponds differentiation of the major arthropod groups. This would certainly strengthen the finding.
The authors do not explain how cytopathic effects (CPE) were characterized. Since they claim CPE manifest in a tissue dependent fashion, they should explain what these effects are. I also think it a weak argument to extrapolate from tissue culture to determine tissue tropism.
Reference 26 does not appear to be a paper but rather a quotation from a paper? This should be corrected by adding proper citation.
Author Response
Response to Reviewer 1 Comments
Point 1: While the title is quite clear, the introduction of the paper is unnecessarily obscure. The third paragraph appears to have been truncated and fails to state the purpose of the paper and the key finding. It should be clearly stated that the study is meant to report on a NOVEL virus (SBV1867-2) and introduce the purpose of the report.
Response 1: We agree with you and have revised the third paragraph as following:
In this study, during a surveillance of viruses carried by sandflies, a novel phlebovirus, sandfly-borne virus 1867-2 (SBV1867-2) was isolated from sandflies collected in Shanxi Province, central China. The virus is a close relative of RVFV based on the the phylogenetic analysis of the RdRp gene of this virus.
Point 2: That this is a rift valley fever like virus found in sandflies (rather than mosquitoes) is an important finding. The authors indicate that sequence difference infer that the divergence of mosquito and sandfly phleboviruses “have taken a long time to evolve”. It would be nice if they set a clock to this divergence and/or look at the chronology of bunyavirales divergence to determine whether it is likely that this divergence corresponds differentiation of the major arthropod groups. This would certainly strengthen the finding.
Response 2: Thank you for your comments and suggestions. In view of that this manuscript mainly reports the isolation and phylogenetic analysis of the novel phlebovirus from sandflies, in which the results are relatively limited (which is one of the reasons why this article is submitted in the format of a communication), the molecular dating of the novel virus and other phleboviruses and chronology of bunyavirales divergence in arthropod groups will be studied in our following article. Our research team will collect sandflies and mosquitoes in the same area where the virus was isolated in 2021 and research the relationship between the virus and blood-sucking insect vectors.
Point 3: The authors do not explain how cytopathic effects (CPE) were characterized. Since they claim CPE manifest in a tissue dependent fashion, they should explain what these effects are. I also think it a weak argument to extrapolate from tissue culture to determine tissue tropism.
Response 3: Thank you for your suggestion. We observed no cytopathic effects in different cell lines infected with SBV1867-2, including of BHK-21, Vero, Vero E6 and C6/36. However RVFV caused CPE in BHK-21, Vero, and Vero-E6 cells, while it didn’t cause significant CPE in C6/36 cells. The infectivity of SBV1867-2 to the above cell lines is different to that of RVFV. It seemed that “infectivity” is more suitable than “tropism” in our manuscript. So we replaced “tropism” with “infectivity” in the revised version.
Point 4: Reference 26 does not appear to be a paper but rather a quotation from a paper? This should be corrected by adding proper citation.
Response 4: Ref 26 is an ICTV Proposal. The format was not correct in the original version. We have corrected it in the revised version as following:
Marklewitz, M.; Palacios, G.; Ebihara, H.; Kuhn, J.; Junglen, S. Create four new genera, create seventy nine new species, rename/move seven species, rename/move three genera and abolish one genus in the family Phenuiviridae. In order Bunyavirales; ICTV: Berlin, Germany, 2019.
Reviewer 2 Report
The authors reported a novel phenuivirus from sand fly captured in China, and the virus was most related RVFV which cause severe illness in human. The manuscript is well written, and the interpretation of their results is fair. The results are interesting and make us occur a new question how the virus is related to human and animal public health. I consider the results can be shared with scientific communities through the journal. I hope following my comments help the manuscript to be published in this journal.
Comment#1:
The authors described GPC encoded in M segment devided into three proteins of Gn, Gc, and non-structural protein NSm. In common, phenuivirus and other buniyavirus encode polyprotein G in M segment, and the polyprotein cleaved by particular cellular protease, such as signalase, furin or SKI-1/S1P. The authors should describe what protease recognition sites the progenitor protein GPC had to produce each viral proteins. In addition, there are a gap between NSm and Gn in the GPC, which means the amino acid sequences can be the forth viral protein. The author should discuss on the amino acid sequences between NSm and Gn can be a functional viral protein(s). If the authors could examine the nature of the amino acid sequences, or they tried homology search on the sequences, please provide their results in this manuscript.
Author Response
Response to Reviewer 2 Comments
Point 1: The authors described GPC encoded in M segment devided into three proteins of Gn, Gc, and non-structural protein NSm. In common, phenuivirus and other buniyavirus encode polyprotein G in M segment, and the polyprotein cleaved by particular cellular protease, such as signalase, furin or SKI-1/S1P. The authors should describe what protease recognition sites the progenitor protein GPC had to produce each viral proteins. In addition, there are a gap between NSm and Gn in the GPC, which means the amino acid sequences can be the forth viral protein. The author should discuss on the amino acid sequences between NSm and Gn can be a functional viral protein(s). If the authors could examine the nature of the amino acid sequences, or they tried homology search on the sequences, please provide their results in this manuscript.
Response 1: Thank you for your professional comments and suggestions.
This manuscript mainly focuses on the isolation and phylogenetic analysis of the novel phlebovirus from sandflies. Based on the phylogenetic analysis of the RdRp gene, the virus is a virus similar to the Rift Valley fever virus. In view of that the results are relatively limited, which is one of the reasons why this article is submitted in the format of a communication, the more analysis of the polyprotein G in M segment will be dealt with in detail in our next work. Our research team are collecting more similar phleboviruses in the local area.